# Southern Madagascar, polycrisis and project failures: A scoping review

**Léo Delpy** [1,2]*, **Claire Gondard Delcroix** [2,3], **Maxime Galon** [2], **Benoît Lallau** [4], **Isabelle Droy** [3,5]

1 CLERSE—University of Lille, Lille, France, 2 BSE—University of Bordeaux, Bordeaux, France, 3 UMI Source, IRD, Versailles Saint Quentin-en-Yvelines, Boynton Beach, France, 4 CLERSE—Sciences Po Lille, Lille, France, 5 LAM—Sciences Po Bordeaux, Bordeaux, France

* leo.delpy@univ-lille.fr

**Data Availability Statement:** All relevant data are within the manuscript.

**Funding:** The CapSud research project, "Development in the Grand South of Madagascar: Some lessons from 30 years of development

## Abstract

The southern region of Madagascar experiences a series of crises related to agro-climatic, nutritional, security, institutional, and political conditions despite the presence of numerous development aid projects over several decades. To understand this apparent paradox, this scoping review examines 63 peer-reviewed and grey literature studies in both French and English from 1990 to 2023, focusing on project failures in the southern region of Madagascar. The article makes two main contributions. Firstly, in terms of methodology, it presents an original approach to conduct a scoping review in a geographical area characterized by the presence of numerous development players and a low number of scientific articles. Secondly, it represents the very first article to offer a synthesis of the literature analyzing development failures in southern Madagascar. It thus appears that the equilibrium of maldevelopment in southern Madagascar is rooted in the systemic interaction between agri-environmental tensions, the failures of the state and aid, and the inadequate consideration of socio-anthropological dimensions and gender relations.

## Introduction

The southern region of Madagascar is marked by a series of crises related to agro-climatic, nutritional, security, institutional, and political conditions. These crises occur within a context of weakened kinship structures and multiple institutional failures accumulated over several decades: underinvestment in infrastructure and public services, poor governance, and a deteriorating security situation leading to decapitalization. The chronic succession of crises exacerbates the vulnerability of the population [1–3]. In 2018, 42% of children under the age of 5 suffered from stunted growth (chronic malnutrition); this proportion rose to 50% in the three southern provinces of the country [4]. The south of Madagascar faces major challenges in relation to the Sustainable Development Goals (SDGs), in particular the goal one on eradicating poverty and the goal two on eradicating hunger and achieving food security, improved nutrition and sustainable agriculture [5].

Faced with these recurring crises, a portion of Southern Madagascar has gradually shifted towards a situation of emergency aid, characterized by the massive presence of international

projects. Bibliographic capitalization study," was conducted with funding from the Delegation of the European Union to Madagascar (DUEM) through the European Development Fund (FED) allocated to the program "Support for Agriculture Financing and Inclusive Value Chains in the South of Madagascar" (Afafi-Sud). The funders had no role in study design, data collection and analysis, decision to publish, or preparation of the manuscript. This article reflects the authors' perspectives, and the European Commission is not responsible for the use that may be made of the information it contains.

**Competing interests:** The authors have declared that no competing interests exist.

organizations [6], coupled with a withdrawal of public organizations [7]. The numerous failures of these projects, interventions, and public policies have earned the region the nickname "cemetery of projects" [2]. The region exemplifies a situation of maldevelopment based on the overlay of projects, ranging from short to long term, replicated without caution in singular geographies and contexts [2,3,8,9].

The remnants of projects are clearly visible in the landscape of southern Madagascar. At the entrance to Amboasary, for example, you can see the ruins of a project-built market that was never used. It's not uncommon to see abandoned solar pumps or impluviums reduced to puddles of stagnant water, unusable even for animal consumption. In many village where projects have been carried out, there are local organizations set up to implement the projects, but they have remained empty shells, with no resources and no action since the projects came to an end. The main development players in southern Madagascar, concerned about the problem, have put forward a set of explanatory elements that overlap with the literature on project failures in other geographical areas.

The main factors identified to explain project failure are (i) climate change-related constraints, which increase pressure on water resources, environmental resources, and harvests; (ii) the social constraints of a society marked by multiple taboos and strong social norms; (iii) the region's landlocked status, which makes it difficult to connect to markets; (iv) the region's under-administration; and (v) the failure to account for local social structures and prevailing norms. Additional factors include the limitations of project logic, such as the multiplication of players and a lack of coordination, which is exacerbated by the project timeframes that are too short; the need to achieve implementation objectives (disbursement rate, equipment rate) without always having a coherent project logic; and the weak presence of the State.

Although the literature on development projects failures in southern Madagascar is abundant, there is no synthesis of this work.

Despite numerous interventions in southern Madagascar, to our knowledge, there is no synthesis of these works. This article aims to fill this gap by proposing a scoping review [10] of studies in southern Madagascar to examine the socio-institutional, socio-economic, anthropological, nutritional, climatic, and historical dynamics related to recurring project failures in the southern region of Madagascar. Considering the diversity of resources available on our study topic (e.g., scientific articles, institutional reports, videos, policy notes, etc.), we propose adopting an original methodological approach in collecting bibliographic resources [11]. Firstly, we conducted a scoping review of the scientific literature using the Pubmed, Scopus, and Web of Science databases. Given the limited number of identified articles, we supplemented this initial approach with a complementary second one. To obtain these other references, we approached the 27 organizations of the Humanitarian, Development, Peace Nexus (a coordination group of national and international organizations working in southern Madagascar). This complementary approach allowed us to build a robust and diverse corpus of documents.

The main contributions of this article are twofold. Firstly, it provides an overview of existing studies in southern Madagascar. In this regard, it synthesizes the primary challenges present in the southern region of Madagascar, including institutional, climatic, and security issues. Secondly, from a methodological standpoint, this article introduces an original approach in implementing the scoping review. It represents a novel contribution to gather references from grey literature. The combination of an approach focused on field actors (Non-Governmental Organizations, Government, Civil Society, etc.) and a conventional approach concentrated on academic literature allows for a comprehensive understanding of the studies conducted in southern Madagascar.

## Material and methods

### Search strategy

We conducted a scoping review following the PRISMA-ScR (Preferred Reporting Items for Systematic Review and Meta-Analyses–Scoping Review) guidelines [10]. This article provides a significant methodological contribution by introducing a novel approach for systematically collecting documents from both scientific and grey literature. To achieve this, we adapted the recommendations outlined by Jenkins et al. (2022).

**Scientific literature.** Initially, we conducted a review of the scientific literature [12]. To achieve this, we performed a search on Scopus, PubMed, and Web of Science databases. The literature search focuses on scientific literature, in French and English, published between 01/01/1990 and 30/05/2023. The formulation of the search equation and, more broadly, our search strategy were validated by a librarian. In the search equation, we distinguished four concepts: (i) geographical limitation, (ii) development projects, (iii) socio-institutional factors, and finally, (iv) the thematic criterion that encompasses issues related to nutrition, climate, agriculture, and livestock. See supplementary materials and Table 1 for details.

**Grey literature.** Given the limited number of identified scientific references, we propose to carry out an additional search focusing on grey literature [11]. Firstly, we reached out to the 27 organizations affiliated with the Humanitarian, Development, Peace Nexus, requesting documents related to the studied issues. To achieve this, we initiated contact through email, followed by telephone follow-ups. Furthermore, we engaged with these organizations during regular meetings organized by the Nexus in Antananarivo. Secondly, a search for references was conducted using the Horizon database of the French National Research Institute for Sustainable Development (IRD). This database, facilitating the digitization and archiving of non-digitized articles, played a crucial role in enhancing our understanding of socio-anthropological dynamics.

**Limitations.** This review only considered studies in the French and English languages, and publications in the Malagasy language were not included. However, French is prominently used by development and governmental agencies, and while some publications are in English, hardly any are in Malagasy. This study focuses on the three regions of southern Madagascar, so articles highlighting development issues across the entire country may not be integrated into this review unless they specifically address the three regions. The collection of grey literature focused on organizations affiliated with the Humanitarian, Development, Peace Nexus. Therefore, non-member organizations operating in the greater south were not approached.

### Study selection

All collected records underwent a screening process applying four inclusion criteria and four exclusion criteria (see Table 2). This selection was performed based on the examination of

**Table 1. Terms for search in bibliographic databases.**

| Geographical Criteria | Development Project Criteria | Socio-Institutional Factors Criteria | Thematic Criteria |
|---|---|---|---|
| Madagascar; South; Sud; Androy; Atsimo-Andrefana; Atsimo Andrefana; Atsimo; Andrefana; Anosy | Development; développement; fail; échec; project; projet; prog | institution*; cultu*; socio*; anthropo*; system*; politi*; state; état; gouv*; crim* | climat*; "changement climatique"; "climate change"; "réchauffement climatique"; "dérèglement climatique"; "global warming"; "effet de serre"; "greenhouse effect"; sécheresse; drought; météo; weather; environnement; environment; ressource*; naturel*; natural; deforestation; déforestation; "ressource naturel"; "natural ressources"; eau; water; aqua*; trawl*; agr*; pêche; fish*; élevage; rear*; rais*; breed*; nutrition*; alimen*; kéré; kere; sécurité; security; "sous-alimentation"; undernourishment; "urgence alimentaire"; "food emergency"; genre; gender; wom*; fem* |

**Table 2. Inclusion and exclusion criteria.**

| Inclusion | Exclusion |
|---|---|
| Records addressing socio-institutional, economic, or environmental factors in the analysis of project failures. | Records addressing other thematics. |
| Records studying at least one of the three regions of the Greater Southern Madagascar. | Records not studying at least one of the three regions of the Greater Southern Madagascar. |
| Records published after the year 1990. | Records published before the year 1990. |
| Record is in English or French. | Record is not in English or French. |
| Records describes a development or emergency project | Records do not describes a development or emergency project |

titles and abstracts. Subsequently, we evaluated the full texts of the extracted references using the same criteria as in the initial screening phase. In the event of disagreement in the selection of a record, resolution was achieved through discussion and consensus among the authors [13].

## Data extraction

Based on the selected records, we proceed with the extraction of data used in this article. We employ 32 variables designed to collect information on the themes studied in the references (e.g., climate change, nutrition, project failure, etc.), the methodological framework of the reference, a critical analysis of the record, as well as the main findings identified in the document (see Table 3).

## Results

The scoping review resulted in the collection of 63 records. Only 10 scientific articles were included in our selection, constituting less than 15% of the total records. It is noteworthy to highlight the limited number of scientific articles addressing the studied topics in our scoping review. Additionally, we retained 53 grey literature references, accounting for 85% of the total selected records. This stage involved the collection of 26 institutional reports, 12 non-peer-reviewed articles, 1 book, 3 theses, and 11 evaluations conducted by NGOs. The detailed process of article and document selection is outlined in Fig 1.

Regarding the analysis of the results, we chose to examine the main themes prevalent in the collection of records: gender [14–22], anthropological factors [1,6,2,9,15,21,23–40], climatic factors [1,14,25,27,41–48], institution [2,7,14,15,18,20,25,33–38,40,43,48–56], technical project management [2,6,9,20,25,30, 31,39,41,49,56–64], human and technical capital [14,16,22,25,26,27,42,43,46,47,60,62,64–68], and finally, nutrition and food insecurity [8,14,16,18,21,22,24,37,43,46–48,60,66,69] (see Table 4). Several results allow us to understand these themes in southern Madagascar. In the selected scientific articles, agriculture (78%), natural resources (70%), and socio-anthropology (74%) are the most frequently addressed themes

**Table 3. Variables used for the characterization of data extraction.**

| Category | Subcategories |
|---|---|
| Basic information | Authors, Year of Publication, Title, Journal, Keywords, Abstract |
| Critical Analysis | Study Design/Methodology, Methodological Aspects |
| Results | Results and Findings |
| Thematic Focus | Agriculture, Climate change, Development Project, Health, Nutrition, Socio-Anthropology, Project Failure, Emergency Project, Water |

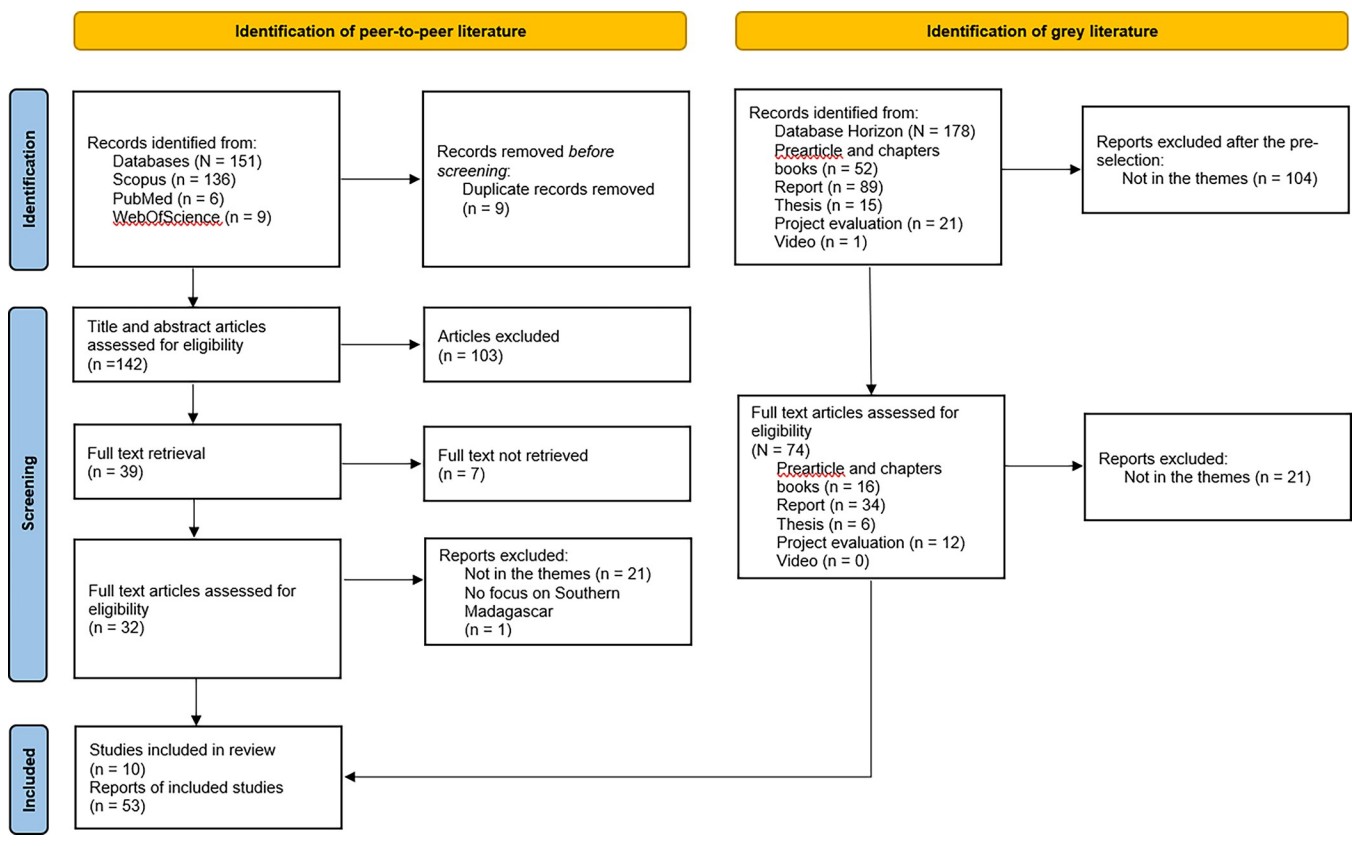

**Fig 1. Flowchart PRISMA.**

within the scope of our scoping review. In grey literature documents, the predominant topics are climate (47%), water (49%), and agriculture (55%).

Concerning project failures, the collected data highlights several explanatory elements. Firstly, 26% of the references address or refer to the lack of consideration for the local context in the implementation of development projects. 17 documents emphasize the importance of tradition, 21 documents highlight local power, and 19 documents discuss socio-economic dynamics. The development of cash transfers is a good illustration of the lack of integration of local contexts in the implementation of projects [61]. The Fiavota social assistance programme in the region is a good illustration of this problem. The study carried out by Gondard Delcroix et al highlights a number of problems, including biased targeting of beneficiaries, communication problems, internal arrangements, and the transparency of the project development process [54]. The lack of contextualisation is also apparent in other development programmes in southern Madagascar, such as the Integrated Food Security Phase Classification [8].

Secondly, 17% of the references address or refer to the colonial past as a potential factor or origin in project failure. These documents underscore the role of the colonial past in regional and national governance processes or environmental crises (e.g., eradication of the cactus, introduction of the cochineal, etc.). Thirdly, 15% of the references address or refer to the lack of infrastructure to explain project failure. The geographical isolation of the region is explained, in particular, by the lack of infrastructure in the region (e.g., roads, telecommunications, etc.). Many programmes focus on specific sites, depending on their accessibility in relation to a number of criteria, such as proximity to a road [14,25,43] or the safety [7,15,48] of the area.

**Table 4. Synthetic table of results.**

| Thematic | Results | References |
|---|---|---|
| Gender | The Essential Consideration of Gender in Development Projects | 14; 15; 16 |
| | Traditionally, Women Are Excluded from Decision-Making Processes | 17; 15; 18; 19; 20 |
| | Low and Unequal Competence and Education among Women Across Geographical Zones | 21; 22 |
| | Strongly Gendered Cultural Norms to the Detriment of Women | 17; 15; 18; 19; 20 |
| Anthropological Factors | Importance of the local context (tradition, local authorities, economic situation, etc.) in setting up projects. Risk of upsetting the social balance | 23; 9; 24; 25; 26; 15; 27; 28; 29; 3; 30; 31; 32; 33; 21; 34; 35 |
| | Importance of the French colonial past (mistrust of foreigners, introduction of cochineal, eradication of cactus. . .) | 36; 9; 37; 24; 26; 6; 2; 1; 33; 38; 39 |
| | Distrust Towards Foreigners | 6; 2 |
| | Lineage tradition sometimes hinders the implementation of projects (particularly with regard to local participation). | 30; 31; 40 |
| Climate Factors | Lack of information on climate impacts | 41; 42; 43 |
| | Vulnerability to climate change | 25; 44; 45; 27; 1; 46 |
| | Drought | 43; 14; 47; 45; 48 |
| Institutional Factors | Lack of infrastructure (landlocked, water, roads, etc.) | 49; 43; 25; 14; 50; 51; 18; 48; 52; 20 |
| | Legal conflict between customary and national law (insufficient consideration of customary law in land law + impact on production due to protected areas) | 36; 33; 34 |
| | State weakness | 53; 54; 37; 7; 50 |
| | Insecurity (Dahalo, Malaso and during the *kéré* period in particular) | 7; 15; 48; 40; 38 |
| | Lack of involvement of the population in the decision-making process (due in particular to the importance of lineage tradition). | 55; 2; 20; 40; 56 |
| Technical Project Management | Project duration too short | 41; 9; 25; 6 |
| | Few structural projects (assistance projects predominate) | 49; 6; 57; 56; 39 |
| | Criticism of the participatory approach: projects do not fit in with the way the community operates and are not adapted to local skills. | 58; 30; 31; 39 |
| | Insufficient project management (communication, scope and targeting) | 59; 60; 61 |
| | Lack of coordination between project players | 62; 6; 63; 56 |
| | Lack of involvement of local players (poor appropriation of projects by local players) | 2; 20; 56; 64 |
| Human and Technical Capital | Lack of production capital (human, financial and material) | 42; 43; 26; 27; 60; 22 |
| | Impact of climate change on production | 25; 47; 27; 46; 64 |
| | Low adoption of modern production techniques (importance of local needs) | 65; 43; 27; 16 |
| | Weak food production (agroecology recommended) | 43; 14; 66 |
| | Importance of livestock farming (off-farm income during the lean period and resilience mechanism) | 62; 47; 16 |
| | Effectiveness of simple development solutions | 67; 68 |
| Nutrition and Food Security | Persistent food insecurity (*kéré*) | 43; 25; 14; 18; 48; 46; 38 |
| | No comprehensive strategy to combat food insecurity | 69; 8; 48 |
| | Malnutrition due to periods of drought | 43; 14; 47; 48 |
| | The importance of women's role in food security | 14; 47; 16; 18; 21; 22 |
| | Importance of malnutrition during lean periods | 46; 66; 60 |

Lastly, the data collection reveals other factors explaining the challenges of aid projects: the lack of integration of gender issues in project implementation (5% of references mention or address this factor), the significance of lineage tradition in Malagasy society impacting project implementation and community participation in decision-making (5%), the impact of climate change, especially drought, and the weakness or absence of the state in development project implementation (10%), insecurity through cattle theft and violence during periods of *kéré* (8%), the lack of genuinely structural projects (8%), a lack of coordination among development

actors (6%), a shortage of funds (10%), and finally, acute malnutrition, mainly attributed to drought and lean periods (11%).

## Discussion

This article provides two major contributions to the understanding of development failures. Firstly, methodologically, this article introduces an original approach to conducting a scoping review in a geographical area characterized by numerous development actors and a low number of scientific articles. The data collection combines a conventional approach (collecting scientific articles) with an innovative field approach (collecting records from stakeholders in southern Madagascar). Secondly, this article synthesizes the diversity of works present in southern Madagascar (e.g. NGO reports, scientific articles, awareness videos, doctoral theses, etc.) on various themes characteristic of development issues in the southern regions of Madagascar (e.g., climate change, water, security, agriculture, politics/institutions, natural resources, socio-anthropology, deforestation, health, nutrition, and human capital). It becomes evident that development crises in southern Madagascar are multifactorial and are linked to diverse elements such as climate change, gender inequalities, the organization and structuring of development projects, and institutional imbalances. Fig 2 visualizes the complex system constituting a maldevelopment equilibrium in Southern Madagascar, as revealed by the main findings identified in the literature reviewed in this scoping review.

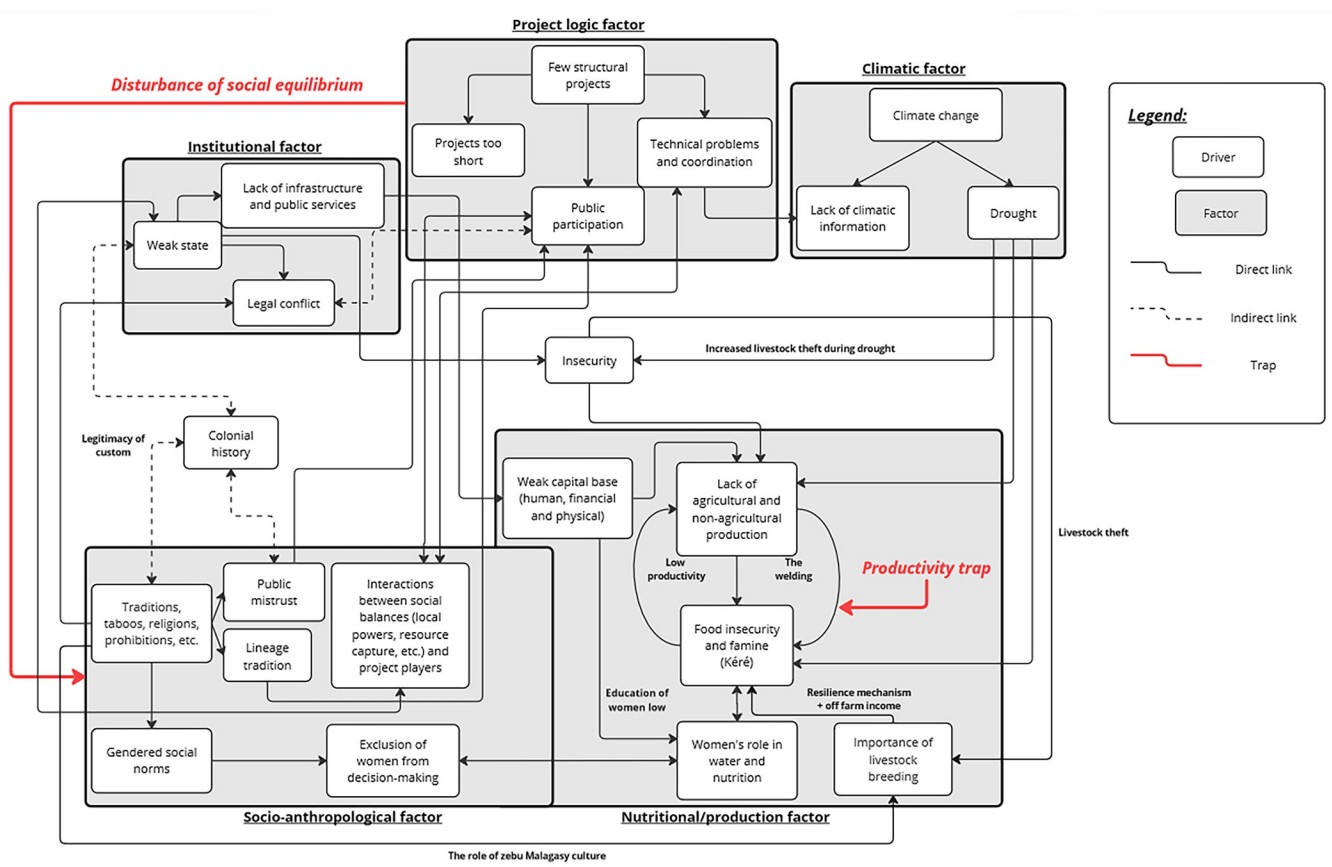

**Fig 2. The complex system constituting a maldevelopment equilibrium in southern Madagascar.** Source: Authors.

## Is southern Madagascar the first zone impacted by climate change?

The south of Madagascar is an area marked by aridity. It is most pronounced in the Tulear region to the west, where annual averages fall below 400 mm of water per year, occurring during a rainy season of less than three months. Rural populations have built pastoral and agricultural systems adapted to this scarcity of rainfall. The danger, however, comes from the extreme irregularity of precipitation, with occasional episodes of drought lasting several seasons. These climatic events impact a context that is particularly economically and socially fragile, subject to repeated food crises (*kéré*) that make it difficult to rebuild production factors [23,51,70]. Thus, while climate change certainly reinforces the already strong agro-environmental constraints of the region, it should not lead to overlooking the central role of economic, security, and socio-institutional factors in explaining the food insecurity in southern Madagascar: political or health crises (such as Covid), deterioration of livelihoods exacerbating chronic poverty, failures of services (insecurity, education, health) and infrastructure [42], more difficult access to water [19], aid failure (lack of coordination among actors [6,9,25,41], project logic [6,56,62,63]. Recently, the *kéré* phenomenon tends to intensify [8,26,38,53] with a growing impact on households' food security [58] and even in the longer term on physical/psychological health. Thus, the malnutrition of mothers has the consequence of the early introduction of foods (other than milk) into the diet of infants [23]. Climate change, to which the south of Madagascar is particularly exposed, risks amplifying extreme phenomena (drought, intensity of cyclones).

## The essential contribution of socio-anthropology

Applying a socio-anthropological perspective to the material of this scoping review highlights two major points of discussion. Firstly, we emphasize the importance of considering the specificities of intervention contexts, a factor often overlooked by projects due to their tendency to duplicate actions considered effective (e.g., cash transfer programs). Specifically, the socio-anthropological approach calls for an examination of local power structures, often decisive in the success or failure of a project [7,23,36,51,58]. Secondly, we note the necessity of incorporating local cultural practices, especially the weight of prohibitions (*fady*) and what is sometimes referred to as "constraints arising from traditions," into projects. However, the risk here is falling into the trap of an "essentializing" approach that tends to consider populations in the South within a static culture incapable of transformations. On the contrary, socio-anthropological studies show us societies in constant evolution, permeable to external influences (especially those of projects), disrupted by contemporary constraints; the opposite of immobile societies [9,29,31,32,54]. What emerges from this scoping review is the necessity of a socio-anthropological analysis in the support and ex-post evaluation of any project, both to increase its chances of success and to have a realistic and unbiased view of the obtained results.

## Strong gender inequalities detrimental to development

Southern societies are lineage-based, highly hierarchical, with strongly differentiated gender roles. The construction of gender relations involves a multidimensional process with symbolic components translated into the political, material, and social frameworks of each society. It determines the rights of individuals based on their gender and position in the lineage, which, however, are not fixed in a society undergoing significant upheavals. The deep-rootedness of gender relations in social structures, with gendered cultural norms disadvantaging women, leads to numerous inequalities, including access to land, credit [15,17–20], and education, which exhibits significant geographic disparities [21,22].

Women's voices in the public sphere are disregarded, resulting in their frequent exclusion or absence from decision-making processes, including those projects primarily concerning them, such as maternal health and nutrition [15,17,18–20]. Despite advancements in understanding the organization of work and the distribution of activities [17,41,57], most agricultural or livestock projects remain blind to gender. Although integrating a gender approach into development activities has become a commitment of development actors and a condition for accessing funding, it is evident that this field is still underexplored [14–16]. It is crucial to better document project contexts from this perspective, which is rarely done before interventions, except in certain NGOs [6,58,70].

## Exploring potential solutions

Southern Madagascar faces numerous social, economic, and climatic risks, yet solutions are possible. At the local level, integration into social norms and social structures is necessary to ensure the sustainability of solutions by fostering ownership, participation, and community buy-in. In this regard, Action Against Hunger's socio-anthropological study [15] underscores that water, due to its symbolic purifying function, is surrounded by many taboos. Moreover, infrastructure management is challenging due to the complexities of property and responsibility surrounding the concept of common goods, necessitating a strong integration of water management mechanisms into local social structures. An example of integration into local norms is provided by a UNICEF sanitation project [35] focusing on CLTS (Community-Led Total Sanitation) and combating open defecation. The project was designed in collaboration with lineage authorities and fokontany authorities to establish a dina (norm set at the fokontany level, enforced by a dina enforcement committee, and subject to sanctions for non-compliance). Agroecology emerges as a viable but demanding pathway for systemic transformation. Agroecological experiments in southern Madagascar appear to enhance population resilience by breaking the aforementioned vicious cycle [71,72]. With this in mind, the agro-ecological block project led by GRET [71,72] has brought about a profound transformation in land use, leading to greater sustainability (reduced water use, agronomic complementarity of crops, harvesting of different species spread throughout the year, windbreaks and fodder for livestock). Supporting the project upstream, during and downstream with anthropological studies and close dialogue with the local population has ensured the sustainable adoption of this innovation. Additionally, they are conducive to better addressing gender issues and highlight technical solutions promoting local autonomy while reducing reliance on imported and expensive equipment and inputs. Finally, the establishment of adaptive social protection mechanisms is a solution developed in the Sahelian region in response to increasing vulnerabilities associated with climate change [73,74]. Establishing such a scheme necessitates a thorough grasp of local contexts, especially the varied landscape of social protection practices, which encompass both informal arrangements (such as interpersonal mutual aid) and formal systems (including social assistance, food aid, etc.) [3,75].

## Conclusion

The documentation on development dynamics in Southern Madagascar is dispersed and often underutilized by development actors. This study has undertaken the task of collecting, storing, and synthesizing scientific works and grey literature from 1990 in the region, providing new perspectives on understanding regional dynamics and their complexities. Several key points of understanding vulnerabilities in the Grand Sud and challenges in implementing development projects, which often exhibit limited sustainability, have emerged. This study opens the door to a systemic analysis of interactions between environmental issues (soil erosion, deforestation,

repeated climatic hazards), deteriorating living conditions (food and nutritional security, access to water), the complexity of cultural norms and gender roles, and the weakness of institutions leading to insecurity and a lack of basic services. An approach that considers this complexity is necessary, and its neglect partially explains the repeated failures of development projects. Addressing these challenges requires a comprehensive and nuanced understanding of the multifaceted factors at play in the Grand Sud region, offering a foundation for more effective and sustainable development interventions.

## Supporting information

**S1 Checklist. Preferred Reporting Items for Systematic reviews and Meta-Analyses extension for Scoping Reviews (PRISMA-ScR) checklist.**
(PDF)

**S1 File.**
(PDF)

## Author Contributions

**Conceptualization:** Léo Delpy, Claire Gondard Delcroix.

**Data curation:** Léo Delpy, Maxime Galon.

**Formal analysis:** Léo Delpy, Claire Gondard Delcroix, Maxime Galon, Benoît Lallau, Isabelle Droy.

**Funding acquisition:** Claire Gondard Delcroix.

**Investigation:** Léo Delpy, Claire Gondard Delcroix, Benoît Lallau, Isabelle Droy.

**Methodology:** Léo Delpy, Maxime Galon.

**Project administration:** Claire Gondard Delcroix.

**Resources:** Léo Delpy, Claire Gondard Delcroix.

**Software:** Léo Delpy, Maxime Galon.

**Supervision:** Léo Delpy, Claire Gondard Delcroix.

**Validation:** Léo Delpy, Claire Gondard Delcroix, Benoît Lallau, Isabelle Droy.

**Visualization:** Léo Delpy, Claire Gondard Delcroix.

**Writing – original draft:** Léo Delpy, Maxime Galon, Isabelle Droy.

**Writing – review & editing:** Léo Delpy, Claire Gondard Delcroix, Benoît Lallau.

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
