## [Decision Letter · Decision Letter 0]

30 Apr 2024

PONE-D-24-05405Southern Madagascar, polycrisis and project failures: a scoping reviewPLOS ONE

Dear Dr. Delpy,

Thank you for submitting your manuscript to PLOS ONE. After careful consideration, we feel that it has merit but does not fully meet PLOS ONE’s publication criteria as it currently stands. Therefore, we invite you to submit a revised version of the manuscript that addresses the points raised during the review process.

We look forward to receiving your revised manuscript.

Kind regards,

George N Chidimbah Munthali

Academic Editor

PLOS ONE

Journal Requirements:

This study was conducted with funding from the Delegation of the European Union to Madagascar (DUEM) through the European Development Fund  (FED) allocated to the program "Support for Agriculture Financing and Inclusive Value Chains in the South of Madagascar" (Afafi-Sud).

Additional Editor Comments:

Dear Authors

Kindly revise the work as suggested by the reviewer

Regards

Reviewers' comments:

Reviewer's Responses to Questions

**Comments to the Author**

1. Is the manuscript technically sound, and do the data support the conclusions?

Reviewer #1: Yes

2. Has the statistical analysis been performed appropriately and rigorously? 

Reviewer #1: Yes

3. Have the authors made all data underlying the findings in their manuscript fully available?

Reviewer #1: Yes

4. Is the manuscript presented in an intelligible fashion and written in standard English?

Reviewer #1: Yes

5. Review Comments to the Author

**Reviewer #1:** The scoping review adds to the body of knowledge and literature on Development evaluation, management and Studies not only for Southern Madagascar but the majority of Low and Middle income countries grappling with project and interventional challenges in Asia, Sub-Saharan Africa and elsewhere across the globe. There are however a number of areas that need to be improved upon before the paper can be deemed publishable as follows:

Background

1. The Background to the scoping review needs to give a precise synopsis of examples of failed projects in Southern Madagascar, perhaps highlighting why the specific projects or interventions failed. Was it something attributable to the nature of the project itself, structural bottlenecks or rather issues of implementation. This early synopsis will present to the reader the status-quo within which project work unfolds and perhaps could provide proper basis for diagnosis or evaluation of challenges.

2. Also the review needs in the introduction to be premised within the scope of Sustainable Development Goals to give it a global appeal. A mention of Goal #1 on ending poverty and Goal # 2 on ending hunger and achieving food security, improved nutrition and sustainable agriculture would suffice.

Methodology

In the two Methodology sections, on Exclusion and Inclusion criteria and on Data extraction, Authors need to include the component on Project evaluation. That is whether the articles considered project evaluation as a component and such should be mentioned as a criteria for inclusion as it could explain failure of the interventions per-se. What indicators were targeted for instance need to be specified.

Discussion

In their discussion, Authors need to furnish a synthesis of what constitutes Development project success first before they dwell on factors or correlates contributing to Development failure. This will help spell out potential diagnostic basis. Authors apart from pointing out pitfalls perpetrating failure of projects in Southern Madagascar perhaps need to suggest potential solutions to the gap. This could be included after the discussion among the additional sections. The recommendation of potential solutions to the gap will resonate with the global nature of the problem. There are diverse Asiatic , sub-Saharan African nations and other low and middle income nations as well as countries around the world with similar contexts, that are grappling with Development implementation issues and national or project failures that would stand to benefit from such analysis. There are failed states where projects have been implemented without any tangible benefits to the poor masses and global resources have been wasted on such. The review adds to the body of knowledge and literature apart from contributing solutions.

6. PLOS authors have the option to publish the peer review history of their article (what does this mean?). If published, this will include your full peer review and any attached files.

Reviewer #1: **Yes: **Dr Marisen Mwale

---

## [Author Response · Author response to Decision Letter 0]

16 May 2024

Comments from Referee

We thank the referee for his/her helpful comments and suggestions, which helped us to improve the paper’s quality. We tried to follow up on suggestions, and below we explain the details of our response on a point-by-point basis. Responses to comments are also available at the end of the document in a more reader-friendly format.

Background

1. The Background to the scoping review needs to give a precise synopsis of examples of failed projects in Southern Madagascar, perhaps highlighting why the specific projects or interventions failed. Was it something attributable to the nature of the project itself, structural bottlenecks or rather issues of implementation. This early synopsis will present to the reader the status-quo within which project work unfolds and perhaps could provide proper basis for diagnosis or evaluation of challenges.

Response

In the introduction, we gave examples of project failures and the main explanations usually put forward by development players. This helps to justify the need for a systematic study to clarify the state of knowledge (See lines 45 to 62). 

We have also added several examples in the results section to illustrate the failures of projects in the south of Madagascar. These examples also enable us to clarify the many reasons for these failures. We have mainly focused on three main reasons highlighted as important by our scoping review.

Firstly, the lack of adaptability to local contexts and the spread of the traveler model. The lack of adaptation of emergency projects to local contexts is widely highlighted in the literature on southern Madagascar and other contexts. With this in mind, we have added elements to highlight this characteristic. One example is cash transfers, which only cover a very small proportion of the population and are difficult to target. 

See lines 159-164 “The development of cash transfers is a good illustration of the lack of integration of local contexts in the implementation of projects [61]. The Fiavota social assistance programme in the region is a good illustration of this problem. The study carried out by Gondard Delcroix et al highlights a number of problems, including biased targeting of beneficiaries, communication problems, internal arrangements, and the transparency of the project development process [54]. The lack of contextualisation is also apparent in other development programmes in southern Madagascar, such as the Integrated Food Security Phase Classification [8].”

Secondly, structural characteristics are also important to explain project failure. The geographical isolation of the region is explained, in particular, by the lack of infrastructure in the region (e.g., roads, telecommunications, etc.). 

See lines 165-172 “Secondly, 17% of the references address or refer to the colonial past as a potential factor or origin in project failure. These documents underscore the role of the colonial past in regional and national governance processes or environmental crises (e.g., eradication of the cactus, introduction of the cochineal, etc.). Thirdly, 15% of the references address or refer to the lack of infrastructure to explain project failure. The geographical isolation of the region is explained, in particular, by the lack of infrastructure in the region (e.g., roads, telecommunications, etc.). Many programmes focus on specific sites, depending on their accessibility in relation to a number of criteria, such as proximity to a road [43 ; 25 ; 14] or the safety [7 ; 15 ; 48] of the area.”

2. Also the review needs in the introduction to be premised within the scope of Sustainable Development Goals to give it a global appeal. A mention of Goal #1 on ending poverty and Goal # 2 on ending hunger and achieving food security, improved nutrition and sustainable agriculture would suffice.

Response

We have added elements relating to the two Sustainable Development Goals mentioned in the introduction. These goals are indeed key elements in understanding the dynamics of vulnerability in southern Madagascar. The region is a good illustration of the difficulties of achieving the goals set in 2015, particularly in areas of extreme poverty such as the south of Madagascar.

See line 35 – 38: “The south of Madagascar faces major challenges in relation to the Sustainable Development Goals (SDGs), in particular the goal one on eradicating poverty and the goal two on eradicating hunger and achieving food security, improved nutrition and sustainable agriculture [5].” 

Methodology

In the two Methodology sections, on Exclusion and Inclusion criteria and on Data extraction, Authors need to include the component on Project evaluation. That is whether the articles considered project evaluation as a component and such should be mentioned as a criteria for inclusion as it could explain failure of the interventions per-se. What indicators were targeted for instance need to be specified.

Response

We have added the elements relating to project evaluation in the two methodology sections. In the section on study selection, we have included the inclusion/exclusion criteria used to select the articles on this basis. In the section on data extraction, we have added elements for characterizing project evaluation. We have added details of the variables used to collect the data for the scoping review. Thus, particular attention was paid to capturing this dimension (project failure) in this article.

Discussion

In their discussion, Authors need to furnish a synthesis of what constitutes Development project success first before they dwell on factors or correlates contributing to Development failure. This will help spell out potential diagnostic basis. Authors apart from pointing out pitfalls perpetrating failure of projects in Southern Madagascar perhaps need to suggest potential solutions to the gap. This could be included after the discussion among the additional sections. The recommendation of potential solutions to the gap will resonate with the global nature of the problem. There are diverse Asiatic, sub-Saharan African nations and other low and middle income nations as well as countries around the world with similar contexts, that are grappling with Development implementation issues and national or project failures that would stand to benefit from such analysis. There are failed states where projects have been implemented without any tangible benefits to the poor masses and global resources have been wasted on such. The review adds to the body of knowledge and literature apart from contributing solutions.

Response

We have added new section in the discussion relating to the success factors of a development project. See section "Exploring Potential Solutions” (Lines 247 – 271). “Southern Madagascar faces numerous social, economic, and climatic risks, yet solutions are possible. At the local level, integration into social norms and social structures is necessary to ensure the sustainability of solutions by fostering ownership, participation, and community buy-in. In this regard, Action Against Hunger’s socio-anthropological study [15] underscores that water, due to its symbolic purifying function, is surrounded by many taboos. Moreover, infrastructure management is challenging due to the complexities of property and responsibility surrounding the concept of common goods, necessitating a strong integration of water management mechanisms into local social structures. An example of integration into local norms is provided by a UNICEF sanitation project [35] focusing on CLTS (Community-Led Total Sanitation) and combating open defecation. The project was designed in collaboration with lineage authorities and fokontany authorities to establish a dina (norm set at the fokontany level, enforced by a dina enforcement committee, and subject to sanctions for non-compliance). Agroecology emerges as a viable but demanding pathway for systemic transformation. Agroecological experiments in southern Madagascar appear to enhance population resilience by breaking the aforementioned vicious cycle [73,74]. With this in mind, the agro-ecological block project led by GRET [73,74] has brought about a profound transformation in land use, leading to greater sustainability (reduced water use, agronomic complementarity of crops, harvesting of different species spread throughout the year, windbreaks and fodder for livestock). Supporting the project upstream, during and downstream with anthropological studies and close dialogue with the local population has ensured the sustainable adoption of this innovation. Additionally, they are conducive to better addressing gender issues and highlight technical solutions promoting local autonomy while reducing reliance on imported and expensive equipment and inputs. Finally, the establishment of adaptive social protection mechanisms is a solution developed in the Sahelian region in response to increasing vulnerabilities associated with climate change [75,76]. Establishing such a scheme necessitates a thorough grasp of local contexts, especially the varied landscape of social protection practices, which encompass both informal arrangements (such as interpersonal mutual aid) and formal systems (including social assistance, food aid, etc.) [3,77].”

In addition, the section on anthropology underlines several solutions. In line with the results identified in the scoping review, it appears that the development of this type of approach provides a precise understanding of the intervention context. This approach, developed in certain projects in the south of Madagascar, offers an interesting solution for adapting projects to local contexts, beneficiaries' expectations and social relationships. Our main recommendation is therefore to promote long, analytical approaches that provide a detailed understanding of contexts in order to avoid problems of implementation in projects. We emphasize this aspect in the discussion 

See lines 216-230 "Applying a socio-anthropological perspective to the material of this scoping review highlights two major points of discussion. Firstly, we emphasize the importance of considering the specificities of intervention contexts, a factor often overlooked by projects due to their tendency to duplicate actions considered effective (e.g., cash transfer programs). Specifically, the socio-anthropological approach calls for an examination of local power structures, often decisive in the success or failure of a project [36,23,58,7,51]. Secondly, we note the necessity of incorporating local cultural practices, especially the weight of prohibitions (fady) and what is sometimes referred to as "constraints arising from traditions," into projects. However, the risk here is falling into the trap of an "essentializing" approach that tends to consider populations in the South within a static culture incapable of transformations. On the contrary, socio-anthropological studies show us societies in constant evolution, permeable to external influences (especially those of projects), disrupted by contemporary constraints; the opposite of immobile societies [54,9,29,1,31,32]. What emerges from this scoping review is the necessity of a socio-anthropological analysis in the support and ex-post evaluation of any project, both to increase its chances of success and to have a realistic and unbiased view of the obtained results."

---

## [Decision Letter · Decision Letter 1]

30 May 2024

Southern Madagascar, polycrisis and project failures: a scoping review

PONE-D-24-05405R1

Dear Dr. PLOSE ONE Chief Editor.

We’re pleased to inform you that your manuscript has been judged scientifically suitable for publication and will be formally accepted for publication once it meets all outstanding technical requirements.

Kind regards,

George N Chidimbah Munthali

Academic Editor

PLOS ONE

Additional Editor Comments (optional):

Reviewers' comments:

Reviewer's Responses to Questions

**Comments to the Author**

1. If the authors have adequately addressed your comments raised in a previous round of review and you feel that this manuscript is now acceptable for publication, you may indicate that here to bypass the “Comments to the Author” section, enter your conflict of interest statement in the “Confidential to Editor” section, and submit your "Accept" recommendation.

Reviewer #1: All comments have been addressed

2. Is the manuscript technically sound, and do the data support the conclusions?

Reviewer #1: Yes

3. Has the statistical analysis been performed appropriately and rigorously? 

Reviewer #1: Yes

4. Have the authors made all data underlying the findings in their manuscript fully available?

Reviewer #1: Yes

5. Is the manuscript presented in an intelligible fashion and written in standard English?

Reviewer #1: Yes

6. Review Comments to the Author

Reviewer #1: Dear Authors

All recommended comments have been addressed satisfactorily as outlined in the comments submitted for the last review. The paper addresses an important component in both health and development studies not only for Madagascar but other low and middle income settings with impact that could be replicable.

Regards,

Dr Marisen Mwale

7. PLOS authors have the option to publish the peer review history of their article (what does this mean?). If published, this will include your full peer review and any attached files.

Reviewer #1: **Yes: **Dr Marisen Mwale

---

## [Editor Report · Acceptance letter]

16 Jul 2024

PONE-D-24-05405R1 

PLOS ONE

Dear Dr. Delpy, 

I'm pleased to inform you that your manuscript has been deemed suitable for publication in PLOS ONE. Congratulations! Your manuscript is now being handed over to our production team.

Kind regards, 

on behalf of

Mr George N Chidimbah Munthali 

Academic Editor

PLOS ONE